# Characterization of polarization states of canine monocyte derived macrophages

**Qingkang Lyu[1,2], Edwin J. A. Veldhuizen[1], Irene S. Ludwig[1], Victor P. M. G. Rutten[1,3], Willem van Eden[1], Alice J. A. M. Sijts[1], Femke Broere[1,4]***

**1** Department Biomolecular Health Sciences, Faculty of Veterinary Medicine, Utrecht University, Utrecht, The Netherlands, **2** Immunology Center of Georgia, Augusta University, Augusta, GA, United States of America, **3** Department of Veterinary Tropical diseases, Faculty of Veterinary Science, Pretoria University, Pretoria, South Africa, **4** Department of Clinical Sciences of Companion Animals, Faculty Veterinary Medicine, Utrecht University, Utrecht, The Netherlands

* f.broere@uu.nl

**Data Availability Statement:** All relevant data are within the manuscript and its Supporting Information files.

**Funding:** The China Scholarship Council (CSC) offered a scholarship with award number

## Abstract

Macrophages can reversibly polarize into multiple functional subsets depending on their micro-environment. Identification and understanding the functionality of these subsets is relevant for the study of immune-related diseases. However, knowledge about canine macrophage polarization is still in its infancy. In this study, we polarized canine monocytes using GM-CSF/IFN- γ and LPS towards M1 macrophages or M-CSF and IL-4 towards M2 macrophages and compared them to undifferentiated monocytes (M0). Polarized M1 and M2 macrophages were thoroughly characterized for morphology, surface marker features, gene profiles and functional properties. Our results showed that canine M1-polarized macrophages obtained a characteristic large, roundish, or amoeboid shape, while M2-polarized macrophages were smaller and adopted an elongated spindle-like morphology. Phenotypically, all macrophage subsets expressed the pan-macrophage markers CD14 and CD11b. M1-polarized macrophages expressed increased levels of CD40, CD80 CD86 and MHC II, while a significant increase in the expression levels of CD206, CD209, and CD163 was observed in M2-polarized macrophages. RNAseq of the three macrophage subsets showed distinct gene expression profiles, which are closely associated with immune responsiveness, cell differentiation and phagocytosis. However, the complexity of the gene expression patterns makes it difficult to assign clear new polarization markers. Functionally, undifferentiated -monocytes, and M1- and M2- like subsets of canine macrophages can all phagocytose latex beads. M2-polarized macrophages exhibited the strongest phagocytic capacity compared to undifferentiated monocytes- and M1-polarized cells. Taken together, this study showed that canine M1 and M2-like macrophages have distinct features largely in parallel to those of well-studied species, such as human, mouse and pig. These findings enable future use of monocyte derived polarized macrophages particularly in studies of immune related diseases in dogs.

201606170114 to Qingkang Lyu. The funders had no role in study design, data collection and analysis, decision to publish, or preparation of the manuscript.

**Competing interests:** The authors have declared that no competing interests exist.

## 1. Introduction

Dogs are considered an attractive pre-clinical animal model for human disease studies due to the natural occurrence of many, inherited, immunological diseases, including lymphomas, osteosarcoma, autoimmune hemolytic anemia and atopic dermatitis [1–3]. Considering the general importance of macrophages in multiple immune diseases, the ability to identify canine macrophages and their functional subsets is a valuable asset for studies in the canine species. To date, characterization of macrophages from different sources such as bone marrow, peripheral blood mononuclear cells, peritoneum and spleen have mostly been studied in mice [4]. However, with the exception of PBMCs, it is difficult to obtain these materials from other mammals like dogs, therefore well-characterized monocyte-derived macrophages could serve as a valuable source of macrophages for relevant *in vitro* studies.

The macrophage as a main member of the monocyte/macrophage phagocytotic system (MPS), plays a crucial role in probably all immune responses, including the regulation of inflammation, homeostasis, and tissue repair [5]. As a group of important antigen presenting and cytokine producing cells, macrophages serve as connectors and regulators between innate and adaptive immune responses, balancing anti- and pro-inflammatory responses [6]. Depending on signals in the local microenvironment, macrophages may polarize into functional subsets [7]. Initially, two main macrophage subsets were identified in mice, namely, classically activated (M1) and alternatively activated (M2) macrophages [8, 9]. In this classical view, M1 macrophages have a fried-egg appearance and produce pro-inflammatory cytokines, such as IL-6, IL-12, IL-23, IL-1β and TNF-α [10]. These cells also express high levels of surface markers such as CD40, CD80, CD86 and MHC II, which increases their antigen-presenting ability, and are associated with a role in a more-general Th1 response. In contrast, M2 macrophages have an elongated spindle-like appearance and produce anti-inflammatory cytokines, such as IL-4, IL-13, IL-10 or TGF-β [11]. They have increased surface expression of CD206, CD209 and CD163 on the cell surface [12], and are functionally characterized by a potent phagocytic capacity, tissue repair and wound healing activity, and associated with a more-general Th2 response [13].

Nowadays it has become clear that these subsets are only artificially defined, extreme, states of the continuous spectrum of all differentiation states that macrophages can have due to their plasticity and that this polarization is actually reversible [14]. Several other differentiation states such as Mox M(hb) and M4 have been defined besides the M1/M2 subsets, while the M2 subset is now divided in several sub-subsets (M2a-c), defined either by their activating ligands and receptors or their production of cytokines [15–17]. Although this labeling of subsets is supposed to provide some clarity, it is obvious that due to plasticity of macrophages, small changes in their microenvironment can have relatively large effects on gene- and protein expression, and thereby functionality, which cannot be captured completely in descriptive labels. However, considering the difficulty of experimentation with macrophages *in vivo*, well defined *in vitro* culture systems providing well defined polarized macrophages are an important asset to predict macrophage function *in vivo*.

Tools available for characterization and use of polarized macrophages in dogs are limited. In 2017 and very recently in 2022, the polarization into M1 and M2 macrophages derived from monocytes was described [18, 19] including microRNA analyses of genes and pathways related to polarization. Canine histiocytic cells (DH82 cells) and monocytes polarized towards two subsets resembling M1 and M2a macrophages, were described in a similar way, but based on a relatively small set of genes and markers [20].

In the present study we describe *in vitro* polarization of canine monocytes into M1 and M2 macrophage, using GM-CSF / IFN-γ + LPS for M1 polarization, or M-CSF/ IL-4 for M2

polarization. The goals of our study were to confirm reproducibility of this methodology to obtain M1 and M2 macrophages in dogs, including appearance of descriptive M1/M2 characteristics. Furthermore, to relate cellular functionality of macrophage M1/M2 subsets as exemplified by phagocytosis to their phenotypes.

## 2. Materials and methods

### 2.1 Canine PBMCs and monocytes isolation

Dog blood samples and buffy coats were obtained from the Companion animal clinics, Faculty of Veterinary Medicine, Utrecht University, under owner's consent. Collection of samples was conducted by professional veterinarians at the department of Clinical Sciences of Companion Animals of Utrecht University. Donors were healthy male and female dog of different breeds between 2 and 8 years old. Blood of >10 dogs was donated, not more often than once every half year. Peripheral blood mononuclear cells (PBMCs) were isolated from canine buffy coats by gradient centrifugation using Histopaque-1077 (Sigma Aldrich, Gillingham, United Kingdom). Briefly, buffy coats were diluted 1:1 in phosphate buffered saline (PBS) and gently layered on top of an equal volume of Histopaque-1077. The gradient was centrifuged for 30 min at 800 ×g at room temperature without braking. The layer of PBMCs located on top of the Histopaque-1077 was collected and washed three times with PBS containing 0.5% fetal calf serum (FCS) and 2 mM EDTA (MACS buffer). Subsequently, monocytes were isolated from the PBMCs by means of MACS using mouse anti-CD14 antibody to label the cells and anti-mouse IgG magnetic Microbeads (Miltenyi Biotec GmbH, Bergisch Gladbach, Germany) using standard protocols.

### 2.2 Culture and polarization of canine monocytes

Isolated CD14$^+$ monocytes were directly seeded into a 24-well plate in 1 ml at a density of $7.5 \times 10^5$ cells/well. Cells were cultured in RPMI-1640 GlutaMAX (Life Technologies$^{TM}$ Ltd., Paisley, Scotland, UK) supplemented with 5% fetal bovine serum (BODINCO B.V., The Netherlands), and 1% penicillin/streptomycin (Life Technologies$^{TM}$ Ltd., Paisley, Scotland, UK) at 37˚C and 5% $CO_2$. To generate monocyte-derived macrophages (MDMs), canine CD14$^+$ monocytes were incubated with canine recombinant GM-CSF (R&D systems, Abingdon, United Kingdom) or recombinant human macrophage colony stimulating factor (M-CSF) (Gibco$^{TM}$, Thermo Fisher Scientific, Carlsbad, USA). The medium was refreshed every 2 days. At day 5, GM-CSF treated monocytes were activated with 20 ng/ml recombinant canine IFN-γ (R&D system, Abingdon, United Kingdom) and 100 ng/ml LPS from *Escherichia coli* O111:B4 (Sigma-Aldrich, Saint Louis, MO, USA) for M1 polarization; M-CSF treated monocytes were activated with 40 ng/ml recombinant canine IL-4 (R&D system, Abingdon, United Kingdom) for M2 polarization.

Control, M0 cells, representing undifferentiated monocytes were derived by culturing monocytes in exactly similar conditions, except that no stimuli (M/GM-CSF, IFN- γ and LPS) were added. M1, M2 polarized macrophages and M0 monocytes were harvested on day 7 for phenotypical characterization, mRNA isolation and phagocytosis assays.

### 2.3 Morphological characterization of canine monocyte-derived macrophages by phase-contrast microscopy

For monocyte-derived macrophages, the morphology of polarized cells and control undifferentiated monocytes was determined in the wells after 3, 5, and 7 days of culture using an inverted phase-contrast microscope ('Eclipse TS100' Nikon Instruments Europe B.V.,

Table 1. List of antibodies used for flow cytometry.

| Antigen | Target species | Clone | Isotype | Dilution | Source |
|---|---|---|---|---|---|
| CD14, vioblue | Mouse anti-human/canine | TÜK4 | Mouse IgG2a κ | 1:100 | Miltenyi Biotec |
| CD11b, biotin | Mouse anti-dog | CA16.3E10 | Mouse IgG1 | 1:100 | P. Moore |
| CD32B+CD32A biotin | Mouse anti-human/canine | AT10 | Mouse IgG1 | 1:50 | Abcam |
| CD40, R-PE | Mouse anti-human/canine | LOB7/6 | Mouse IgG2a | 1:25 | BIO-RAD |
| CD80, FITC | Hamster anti-mouse/canine | 16-10A1 | Hamster IgG2 | 1:100 | BD Biosciences |
| CD86, unconjugated | Mouse anti-dog | CA24.3E4 | Mouse IgG1 | 1:50 | P. Moore |
| CD83, APC | Mouse anti-human/canine | HB15e | Mouse IgG1, κ | 1:50 | Biolgend |
| CD206, APC/Cyanine7 | Mouse anti-human/canine | 15–2 | Mouse IgG1, κ | 1:50 | Biolegend |
| MHC II, APC | Rat anti-canine | YKIX334.2 | Rat IgG2a, κ | 1:50 | eBioscience™ |
| GaM-IgG1-PerCP | Mouse IgG | - | IgG | 1:100 | Santa Cruz Biotechnology |
| streptavidin -PE | - | - | - | 1:2000 | BD Biosciences |

Amstelveen, The Netherlands) and ImageFocus v3.0.0.2. Data were processed using Fiji imageJ. software Afterwards M0, M1 and M2 cells were harvested for further analyses.

## 2.4 Phenotypic characterization of monocyte-derived macrophages by flow cytometry

For flow cytometric analysis, at day 7, undifferentiated monocytes, and MDM-derived M1 and M2 cells were washed with cold PBS and detached by gentle scraping to obtain monocyte-derived macrophages. Cells were transferred into a round-bottom 96-well plate at a density of $2\times10^5$ cells/well. Fc receptors were blocked with 10% autologous dog serum for 30 min to avoid non-specific antibody binding. To determine their phenotype, macrophages were stained for 30 min on ice using antibodies specific for CD14, CD40, CD80, MHC II, CD206, CD83, CD86, CD32, CD11b, or isotype controls (see Table 1). Cells were washed 3 times with FACS buffer (2% FCS in FBS), incubated with secondary or streptavidin conjugated antibodies for another 30 min on ice, washed again, and at least 10,000 events were analyzed using a Cyto-FLEX LX flow cytometer (Beckman Coulter Inc., CA, USA). To exclude dead cells from analyses, the ViaKrome 808 Fixable Viability Dye (Beckman Coulter, Woerden, Netherlands) was used. Data acquired were analyzed with FlowJo Software v.10.5 (FlowJo LCC, Ashland, USA).

## 2.5 RNA isolation from canine MDMs, library preparation and RNA sequencing

After 7 days of polarization as described above, a Qiagen RNeasy mini kit was used to extract total RNA from M0 monocytes, and M1 and M2 macrophages. In addition, newly isolated CD14+ cells and the CD14 depleted (CD14D) fraction of the PBMCs were included as controls. RNA concentration was measured using a NanoDrop-1000 Spectrophotometer (Isogen Lifescience B.V., Utrecht, The Netherlands). RNA quality, library preparation and RNA sequencing were done by Novogene Co., Ltd. More specifically, RNA samples were analyzed on 1% agarose gels and a spectrophotometer (NanoPhotometer,IMPLEN, CA, USA) was used to monitor RNA degradation, contamination, and purity. The RNA Nano 6000 Assay Kit from the bioanalyzer 2100 system (Afilent Technologies, CA, USA) was used to assess RNA integrity and quantity. RNA samples with RNA Integrity Number (RIN) $\geq$ 8 were included in the following analysis.

Following quality control of RNA, up to 1 μg per sample was used for library preparation using NEBNext® UltraTM RNA Library Prep Kit for Illumina® (NEB, USA) according to

the manufacturer's instructions. Briefly, PolyA-containing mRNA was enriched from total RNA using poly-T oligo-attached magnetic beads. RNA fragmentation was performed using divalent cations under elevated temperature in NEBNext First Strand Synthesis Reaction Buffer (5X). First-strand cDNA was synthesized using M-MuLV Reverse Transcriptase (RNase H-) with random hexamer primer. Second-strand cDNA was synthesized using DNA Polymerase I and RNase H. Next, 3' end adenylation and adaptor ligation were carried out. cDNA fragments of 150–200 bp were purified with the AMPure XP system (Beckman Coulter, Beverly, USA) for library fragments. Library quality was evaluated using the Agilent Bioanalyzer 2100 system. Subsequently, cluster generation was carried out on a cBot Cluster Generation System using PE Cluster Kit cBot-HS (Illumina). Finally, the sequencing was performed on a NovaSeq 6000 System (Illumina, San Diego, CA) with paired-end read configuration.

## 2.6 Reads processing and differential expression analysis from RNA-seq data

In order to get clean reads, reads meeting the following conditions were filtered out from the sequenced/raw reads: 1) containing adapters, 2) the quality value of over 50% bases of the read is less than or equal to 5, 3) percentage of non-determined bases is over 10%. Clean reads were mapped to the Canis familiaris.canfam 3.1 reference genome using HISAT2 software v2.0.5 [21, 22].

Differential gene expression between groups was calculated using DESeq2 package from Bioconductor) open source software for bioinformatics) with default parameters in the program R. Genes with read counts < 2 in all samples were filtered out from DEG testing. Genes with |log2 (fold change)| ≥ 1 and an adjusted p value < 0.05 were considered as differentially expressed genes (DEGs). DEGs were assessed through 6 group comparisons: M0 vs M1, M0 vs M2, M1 vs M2, CD14+ vs M0, CD14+ vs M1, and CD14+ vs M2. Heat map generation and cluster analysis were performed using the R package "pheatmap". Volcano plots were generated with "ggplot2" in R packages displayed DEGs for each comparison. Principal component analysis (PCA) was conducted based on normalized Fragments Per Kilobase Million (FPKM) to show the similarity and difference of M0, M1, M2, CD14+ and CD14D populations.

## 2.7 Gene Ontology (GO) enrichment analysis and Kyoto Encyclopedia of Genes and Genomes (KEGG) Pathway enrichment analysis

The annotation of sorted DEGs were performed using the GO and KEGG database. DEGs from each group comparison were used for GO and KEGG enrichment analysis using the R package ClusterProfiler v3.8.1. GO terms enriched within Biological Process (BP), Cellular Component (CC), and Molecular Function (MF) were assessed. In addition, the enrichment of DEGs within KEGG pathways were explored. The differences between groups compared were evaluated using Gene Set Enrichment Analysis (GSEA). GO terms and KEGG pathways with a corrected p value < 0.05 were determined as statistically significant enrichment. The top 20 most enriched GO terms and KEGG pathways in DEGs were shown.

## 2.8 Phagocytosis assay by flow cytometry

Phagocytic capacity of canine MDMs was evaluated using 1 μm crimson carboxylate-modified FluoSpheres fluorescent beads (Life Technologies Corporation, Eugene, USA). Specifically, canine MDMs were co-cultured with FluoSpheres fluorescent beads at a ratio of 10:1 (bead: cell) for 4 h on day 7. Then cells were washed three times to remove non-phagocytized beads. Harvested cells were washed 2 times with PBS containing 5 mM EDTA and transferred into a

96-well U-bottom plate. Next, the cells were stained with ViaKrome 808 Fixable Viability Dye (Beckman Coulter, Woerden, The Netherlands) for 30 min on ice to exclude dead cells from analyses. Afterwards, the cells were washed twice and fixed in 4% paraformaldehyde (Alfa Aesar, Kandel, Germany) for 15 min at room temperature. Finally, the cells (at least 10,000 events) were analyzed in a CytoFLEX LX flow cytometer (Beckman Coulter Inc., CA, USA) using the 638 nm laser and the 660/10 fluorescent channel. Data were analyzed using FlowJo Software v.10.5 (FlowJo LCC, Ashland, USA) and GraphPad Prism 8.3.0 (Graphpad Software LLC., San Diego, USA). The first fluorescent peak showing in the histogram was considered as cells with a single bead. The average bead uptake and the change in bead uptake compared to (M0) undifferentiated monocytes was calculated using formulas described in literature [23].

$$\text{Bead/cell} = \frac{\text{MFI}_{\text{total}}}{\text{MFI}_{\text{1bead/cell}}} \text{ and fold change} = \frac{\text{beads/cell}_{\text{M1 or M2}}}{\text{beads/cell}_{\text{M0}}}$$

## 2.9 Assessment of bead internalization by confocal microscopy

To confirm the internalization of beads in polarized macrophages, CD14+ monocytes were grown on sterilized 12 mm coverslips in a 24-well plate. After 7 days of polarization and activation, 1 µm crimson carboxylate-modified FluoSpheres fluorescent beads (Life Technologies Corporation, Eugene, USA) were added into the 24-well plate at a ratio of 10:1 (beads: cells), and incubated for 4 h at 37˚C. Then macrophages were washed three times with cold Hank's balanced salt solution (HBSS) (Gibco, Paisley, UK) to remove non-phagocytized beads, followed by staining of the macrophage membranes with 2 µg/ml Alexa Fluor 488 conjugated Wheat germ agglutinin (WGA) (Life Technologies Corporation, Eugene, USA) in HBSS for 10 min at 37˚C. Subsequently, macrophages were washed twice with HBSS to remove excess WGA and fixed with 4% paraformaldehyde for 15 min at room temperature. After washing twice, coverslips with macrophages were mounted on polysine slides (Thermo Scientific, Braunschweig, Germany) with FluorSave Reagent (Millipore, San Diego, USA). Finally, uptake and internalization of beads were confirmed using the TCS SPE-II spectral confocal microscope and LAS-AF software (Leica Microsystems B.V., Amsterdam, The Netherlands). Z-stacks were made to determine whether beads were completely internalized. The images from confocal microscopy were analyzed using Fiji imageJ.

## 2.10 Statistical analysis

Statistical analysis was conducted using GraphPad Prism 8.3.0 (Graphpad Software LLC., San Diego, USA). Two-way ANOVA tests with multiple comparisons were used to determine statistical differences. The mixed effects model was used in case of missing values. The mRNA sequencing data were analyzed using R program and appropriate R packages as indicated above. Data were harvested from at least three donor dogs. All independent experiments were repeated at least three times. Each P value is adjusted to account for multiple comparisons. *p<0.05 and **p<0.01 were considered as significant difference and highly significant, respectively.

## 3. Results

### 3.1 Morphologic characterization of canine monocyte-derived macrophages

In order to obtain canine macrophages *in vitro*, CD14+ cells, derived from canine PBMCs were polarized into classically activated M1 (GM-CSF/ IFN-γ + LPS) and alternatively activated M2 macrophages (M-CSF/ IL-4). Phase contrast microscopic analysis of the cultured canine MDMs, showed morphological changes between M0 monocytes and MDM subtypes

M1 and M2 from day 5 onwards (**Fig 1A**). At day 3, all subsets had a small and roundish shape, but at day 5, M1-polarized macrophages started to display morphological changes such as increased size, and a flat, roundish, and amoeboid shape. Some M1 cells adopted a typical well-spread "fried-egg" phenotype (red arrow). A small portion of cells displayed stretched, spindle-like morphology at day 5 in M2-polarized macrophages (blue arrow). At day 7, these morphological changes were more pronounced, M1-polarized macrophages were differentiated into homogenous round cells with a dominant amoeboid and "fried-egg" phenotype, while a large portion of M2-polarized macrophages had obtained an elongated spindle morphology (**Fig 1A**). Flow cytometry analysis showed that M1-polarized macrophages were significantly bigger and more granular than M0-monocytes and M2- macrophages (**Fig 1B and 1C**). Non-viable cells were excluded from analysis (as shown in S1 Fig). Thus, M1 and M2-skewed canine macrophages exhibit clearly distinct morphology.

## 3.2 Phenotypic characterization of canine monocyte-derived macrophages

In order to phenotypically characterize *in vitro* polarized canine MDMs, flow cytometry was performed on day 7 of culture. A set of surface markers related to macrophage activation was selected based on previous studies [24, 25]. As expected, pan-macrophage markers CD14 and CD11b were highly expressed by M0 undifferentiated monocytes, M1 and M2 cells (**Fig 2A**) with significantly increased CD11b expression on M1 and M2 MDMs compared to M0 undifferentiated monocytes. Interestingly, M1 MDMs expressed significantly higher levels of activation markers CD40, CD80, CD86 and MHC II (**Fig 2A**), which are often used as M1 markers in human and mouse studies, compared to undifferentiated M0. MHC II expression had also increased in M2 macrophages. In contrast, in M2 macrophages, surface receptors CD32, CD163 and CD206, CD209 were significantly higher than either M0 or M1 cells, while CD83 was actually reduced in M1 (but not raised in M2 vs M0). These results indicate that, similar to other species, there are distinct differences in expression of specific surface markers between M1 and M2-skewed canine macrophages.

## 3.3 Monocyte-derived M1 and M2 macrophages depict different gene profiles

To investigate the gene expression profiles of undifferentiated monocytes and monocyte-derived M1 and M2 macrophages, mRNA was sequenced. After filtering out genes with read counts < 2, 17,929 genes were identified expressed in the 5 cell types: M0 monocytes, M1 and M2 polarized MDMs and freshly isolated CD14+ and CD14-depleted cells (CD14D) (**S1 Table**). **Fig 3A** shows the distribution of genes among the different subsets in a Venn diagram: M1- and M2- polarized macrophages expressed 10,030 genes in common and 440 versus 816 specific in M1 and M2 macrophages, respectively. In the three-groups-comparison M0 vs M1 vs M2 group, 9,895 genes were expressed by M0, M1 and M2 macrophages. In addition, 244 unique genes in M0 monocytes, 272 unique genes in M1 macrophages and 246 unique genes in M2 macrophages (**Fig 3A**) were expressed. The Venn diagram for a five group comparison including CD14+ and CD14-depleted (CD14D) showed that despite larger partial overlaps each group still has a certain set of group specific genes. To identify differentially expressed genes (DEGs) in the different groups, $p$ <0.05 and |log2(FC)|>1 were set as criteria. Based on these, 13,052 differentially expressed genes (DEGs) were identified (**S2 Table**). Results were plotted as a heat map showing the expression levels of DEGs of each sample (n = 3), from which the expression trend can be clearly seen (**Fig 3B**).

Comparisons between 2 groups reveal interesting details on how polarization of monocytes has changed gene expression of the different MDM subsets. Compared to M0 monocytes,

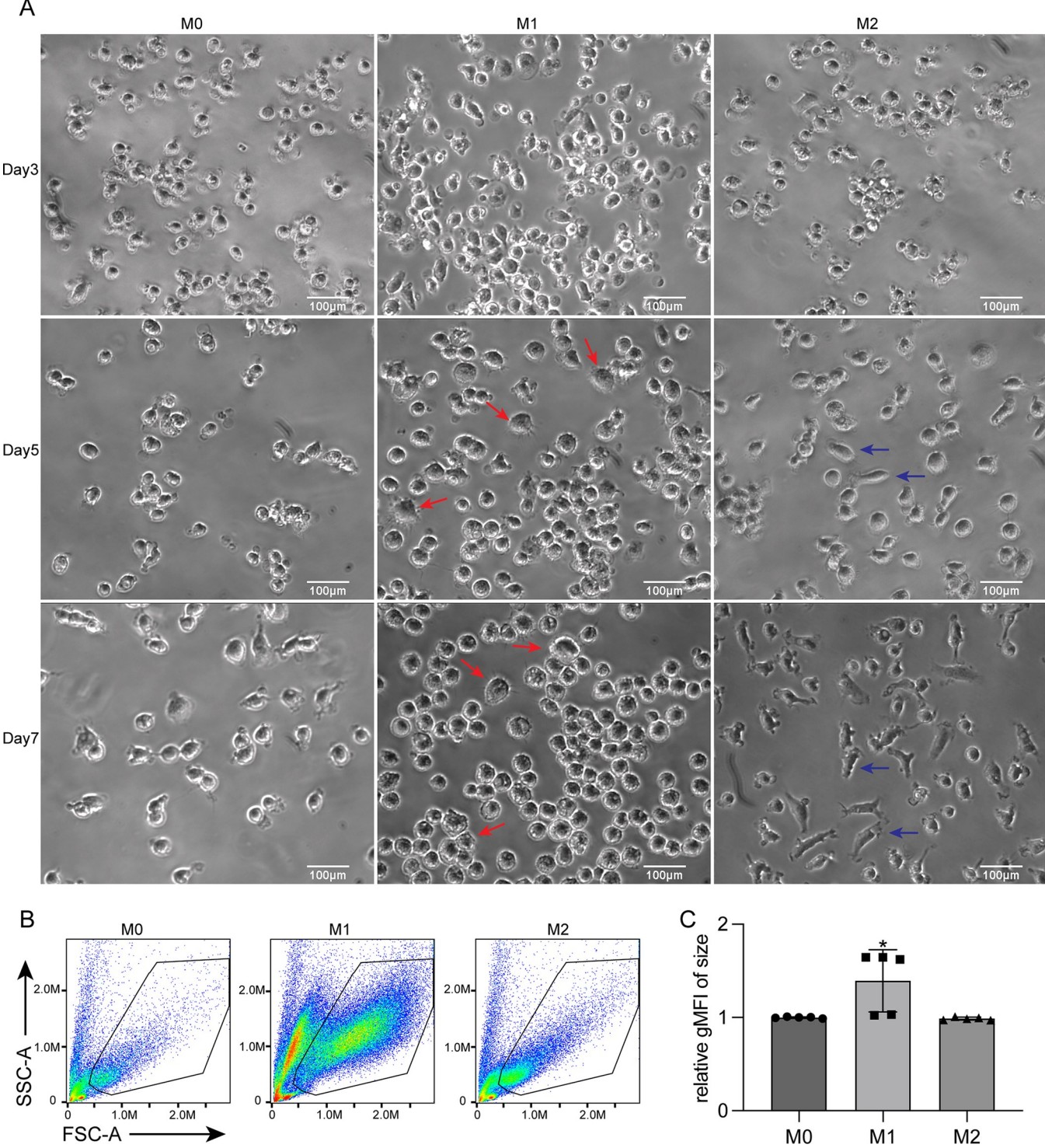

**Fig 1. Morphological changes between canine undifferentiated monocytes and M1 and M2 polarized macrophages.** (A) Representative images showing morphology of undifferentiated monocytes (M0) and monocyte derived M1 and M2 macrophages at day 3, 5 and 7 of culture. M1 macrophages exhibit a large and round shape with amoeboid and flat morphology (red arrow), while M2 macrophages exhibit a stretched and spindle-like morphology (blue arrow). Scale bar = 100 μm. (B) Flow cytometry forward scatter (FSC) profiles indicative of size of monocyte-derived macrophages (C). Geometric mean (gMFI) FSC based size difference between M0, M1 and M2 cells is summarized. Data are presented as mean ± SD, data obtained from 3 different donors measured in triplicates. gMFI: geometric mean of fluorescence intensity, SSC: side scatter, FSC: forward scatter.

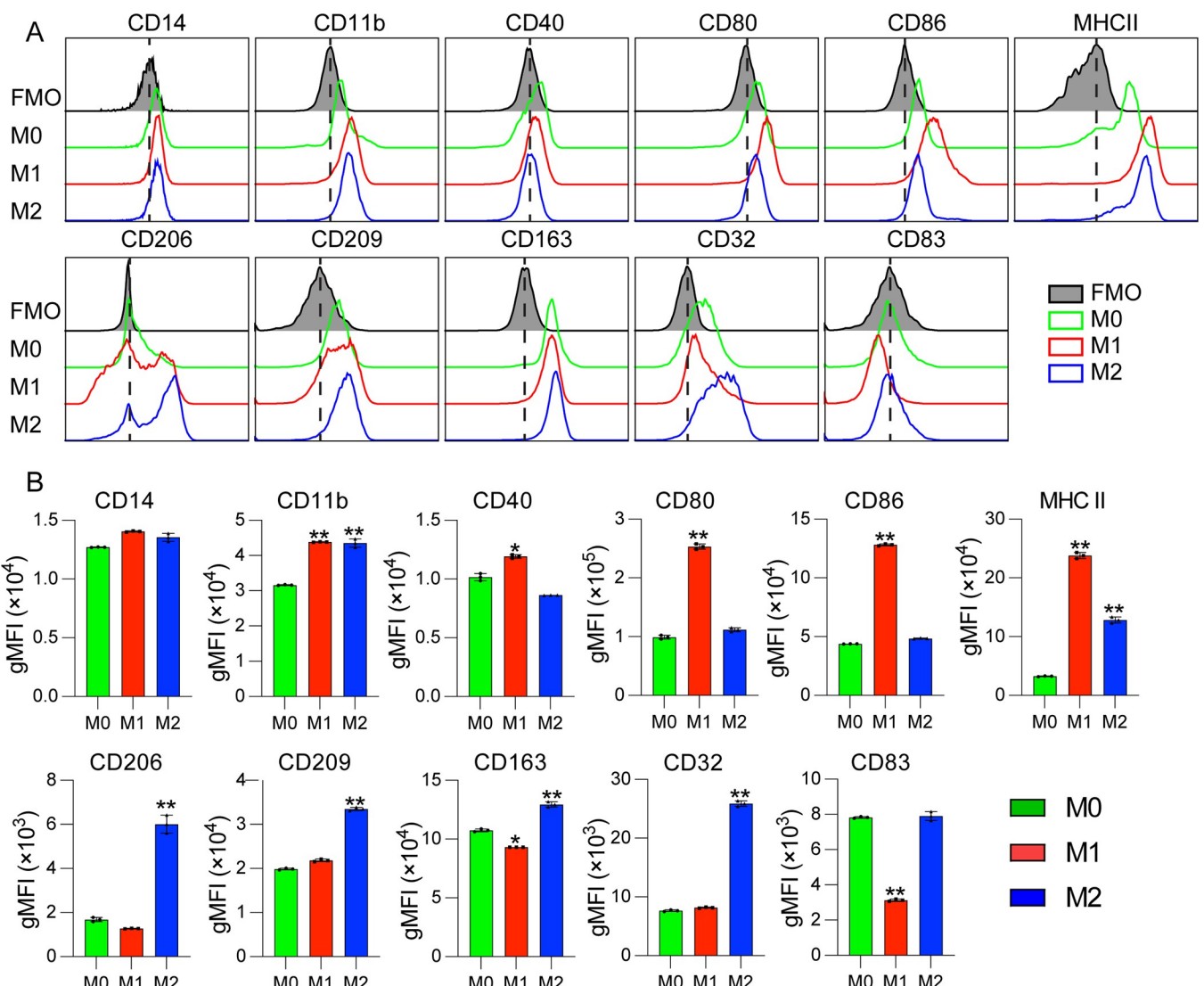

**Fig 2. Flow cytometric analysis of surface markers expressed by canine monocyte derived macrophages.** Undifferentiated monocytes and polarized canine MDMs (M0, M1 and M2) were collected on day 7, stained with a set of macrophage surface markers (see Table 1) and analyzed by flow cytometry. **(A)** Representative histogram obtained from one donor. Each overlaid histogram represents an overlay of the respective monoclonal antibody and fluorescence-minus-one (FMO) controls (shaded black lines) for the indicated marker. Green line: M0, red line: M1, blue line: M2. **(B)** The bar graphs represent the means of geometric fluorescence intensity (MFI) ± SD of positive cells in 3 different donors measured in triplicates. *p* values of M1 and M2 are relative to M0.

most DEGS (3199), 1649 up- and 1550 down regulated, were observed in the M1 group (**Fig 4A**), a much lower number of DEGs, 1882 were observed in the M2 macrophages (901 up, 981 down, **Fig 4B**). A total of 3124 DEGs were found in the M1 vs M2 comparison, with 1688 upregulated and 1436 downregulated genes (**Fig 4C**). In pairwise comparisons including the control CD14+ cells, there were 6,092, 6,283 and 6,330 DEGs between CD14+ and M0, M1 and M2 respectively, with approximately evenly distributed numbers of up- and downregulated genes (S1 Fig). The top 50 up- and down-regulated genes for each pairwise comparison of these 4 groups (M0, M1, M2 and CD14+) are listed in **S3 Table.**

An interesting observation can be made from the large pool of DEGs when specifically the expression of several pro- and anti-inflammatory cytokines is studied (S2 Fig). Clearly, M1 macrophages showed higher expression of M1 marker genes, IL-6, IL-1β, TNF-α, iNOS, LXN,

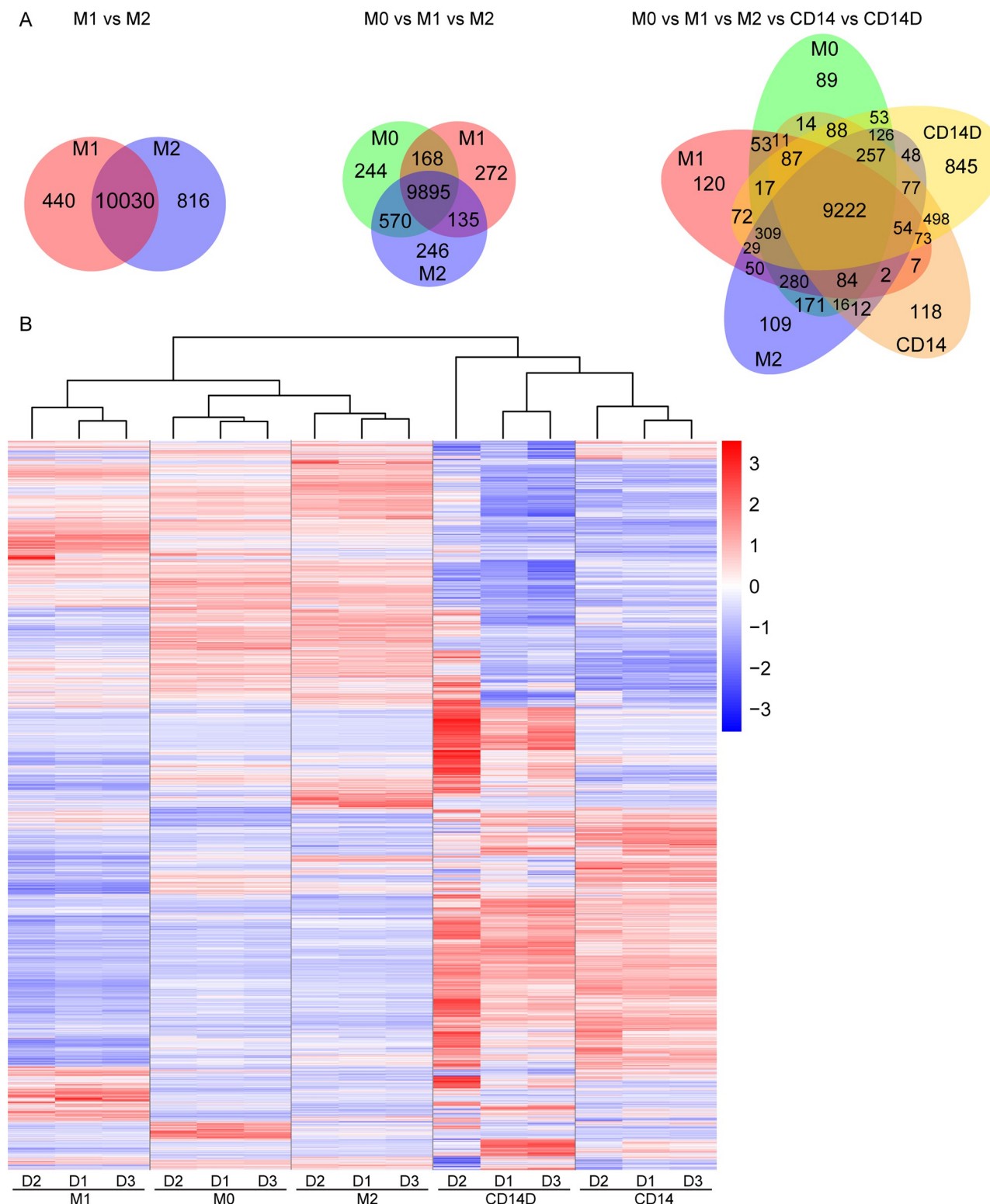

**Fig 3. Gene expression profiles of undifferentiated monocytes (M0), M1, M2, CD14+ and CD14-depleted cells, shown by Venn diagram and heat map.** Venn diagrams showing the distribution of expressed genes in the 5 groups, comparisons are shown for M1 vs M2; M0 vs M1 vs M2; and M0 vs M1 vs M2 vs CD14+ vs CD14D. The number of overlapping and uniquely expressed genes are shown. The overlapping regions indicated the number of co-expressed genes in two or more samples. M0, M1 and M2: different polarization states of macrophages. CD14:freshly isolated CD14+ cells from PBMC. CD14D: CD14+ depleted cell population of PBMC. **(B)** Heat map with cluster analysis showing differentially expressed genes within all

comparison groups. Adjusted p value < 0.05 was used as a cutoff. Each column indicates one sample. Each row indicates a single gene. The color ranges from red to blue represents the expression level of genes from high to low. The log2(FPKM+1) value was used as cut off for cluster analysis. Data were obtained from 3 different donors. FPKM: Fragments Per Kilobase Million.

IL-12A and IL-12B, while CCL2 and COX2 expression were decreased. In addition M2 marker genes MS4A2, TGF-β, IL-10 were shown to be expressed at a higher level in M2 than in M1 macrophages.

To further evaluate clustering of gene profiles, principal component analysis (PCA) was performed. This showed that M2 macrophages clustered with M0 monocytes, while M1, CD14 + and CD14D were clearly distinct from each other (Fig 4D). Overall, the transcriptomic data confirmed that the differentiation protocols used indeed resulted in monocyte-derived M1 and M2 macrophages with significantly different gene profiles and activation patterns.

### 3.4 DEGs of M1 and M2 MDMs were enriched in different GO terms and KEGG pathways

To determine the main changes in biological processes during polarization of MDMs, gene ontology (GO) and KEGG enrichment analysis of all DEGs was performed using the "cluster-Profiler" package in R. The top 20 of most enriched GO terms and KEGG pathways are shown in Fig 5. As expected, the data convincingly show that many of the processes relate to immune function of macrophages. The top 2 GOs for M1 polarization (both vs M0 as M2), within the Biological Process (BP) group are "immune system process" and "immune response", while the KEGG analyses results include functional pathways related to Th17 cell differentiation, TNF signaling and Toll-like receptor signaling. Comparable results were seen in the M1 vs M2 comparison, which also shows the "immune system process" and "immune response" groups in the GO analysis with the highest number of DEGs, while the KEGG analysis includes Th1, Th2 and Th17 cell differentiation and also TNF signaling in the top scoring pathways. Differences between M0 and M2 included other GO terms, enriched DEGs were mainly associated with "GTPase activator activity" and "G-protein coupled receptor binding" in the Molecular Functions (MF) group. The KEGG Pathway analyses did not provide a clear picture how M0 and M2 differ immunologically, since many of the most pathways cannot directly be linked to immune function, with the exception of the chemokine signaling pathway.

Overall, the RNAseq data indicate clearly that also on an gene expression level a substantial difference exists between the macrophage populations. The fact that M1 and M2 differ in many immunological functions, while both M1 and M2 are different from control M0 cells, suggests that M1 and M2 MDMs are distinct populations.

### 3.5 Phagocytic capacity of canine monocyte-derived macrophages

A functional analysis of the MDMs was performed by examining phagocytic capacity of MDMs using crimson carboxylate-modified FluoSpheres fluorescent beads. These experiments showed that all MDMs were able to take up fluorescent beads (Fig 6), while control M0 monocytes co-cultured with beads at 4°C failed to engulf any beads (data not shown). M0 cells and M2-polarized MDM displayed a significantly higher bead-uptake than M1 cells (p < 0.05) (Fig 6A and 6B). On average, M2-polarized MDMs phagocytized 3 times more beads than M1-polarized MDM (Fig 6C). This observation was further qualitatively confirmed by confocal microscopy (Fig 6D), where less fluorescent beads were observed in M1 MDM. The highest concentration of beads was found in the spindle-like shaped cells in the M2 MDM population. Next, the location of fluorescent beads was visualized by re-constructing a 3-D model of

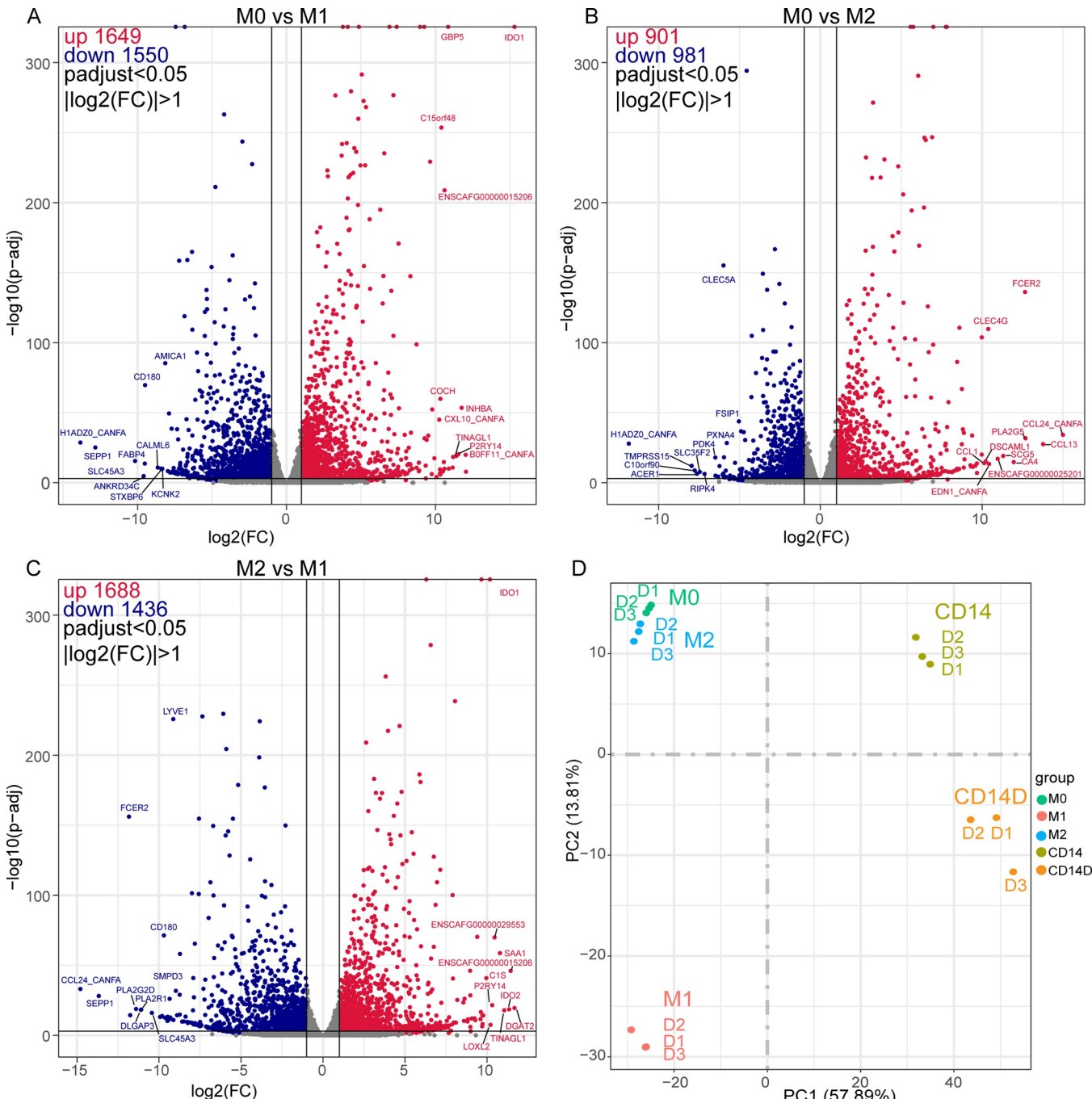

**Fig 4. Polarized macrophage subsets show distinct gene expression profiles.** Volcano plots depicting differentially expressed genes in different comparison groups. **(A)** comparison between undifferentiated monocytes (M0) and M1, **(B)** M0 and M2, and **(C)** M1 vs M2. Each volcano plot was shown as log2 (fold change) (x-axis) vs -log10(p-adj) (y-axis). M0 monocytes are considered as control in **(A)** and **(B)**. M2 macrophages are considered as control in **(C)**. Genes with adjusted p value < 0.05 and |log2(FC)| > 1 are regarded as statistically significantly different. The horizontal line at y ≈ 1.303 indicates $p = 0.05$. Vertical lines at x = 1 or -1 indicate log2(FC) = 1 or -1. Red dots, blue dots and gray dots represent up-regulated, down-regulated, and non-significant DEGs, respectively. The name of top 10 DEGs with $p < 0.001$ and $|log2(FC)| \geq 8$ were labeled out. FC: fold change; DEGs: differentially expressed genes. **(D)** Principal-component analysis of all genes expressed based on FKPM in M0, M1, M2, CD14+ and CD14D.Data obtained from 3 different donors.

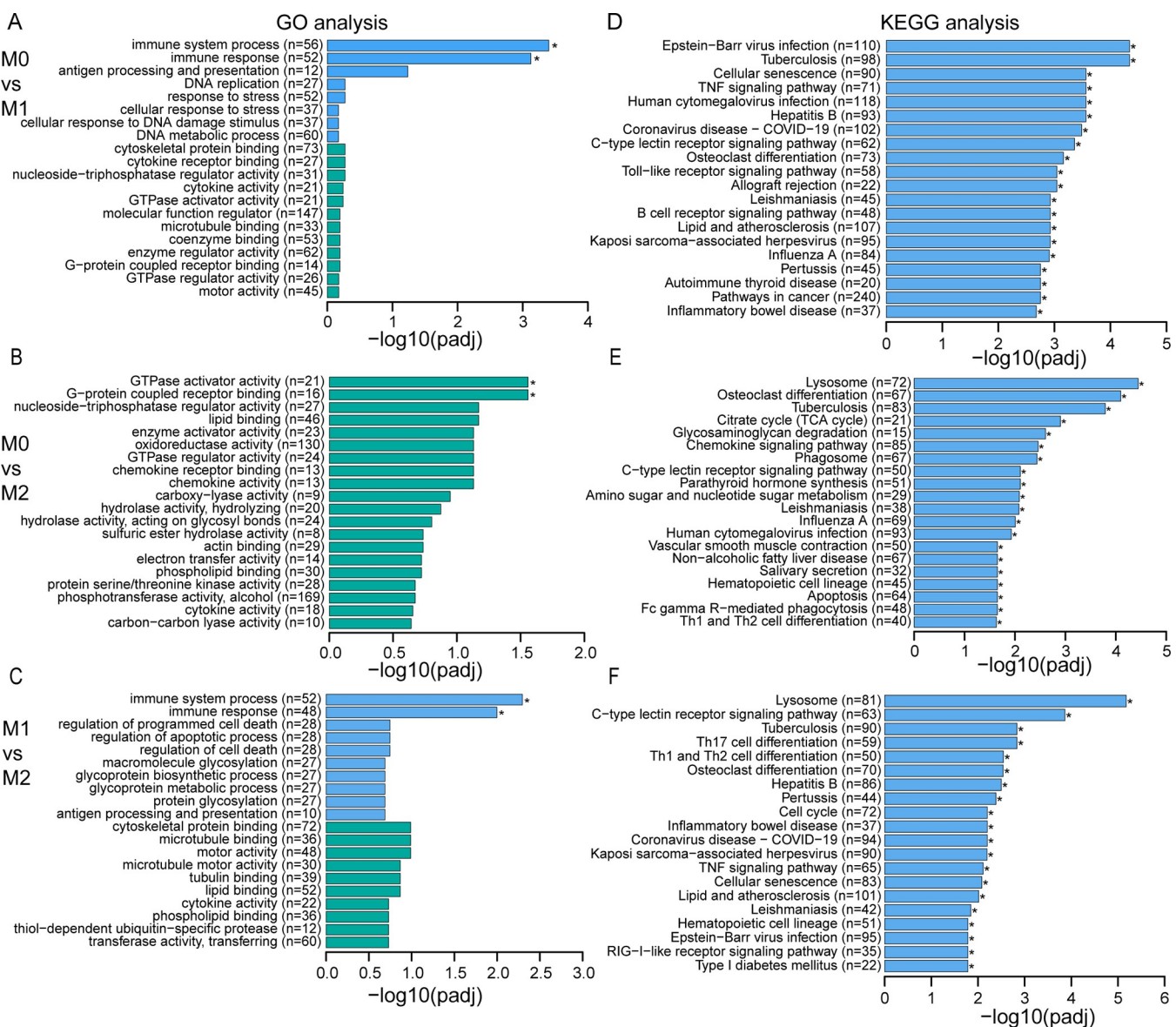

**Fig 5. The GO terms and KEGG pathway enrichment analysis of DEGs between undifferentiated monocytes and M1 and M2 MDMs.** DEGs including both up- and down-regulated genes from each compared group were subjected to GO and KEGG analysis. The top 20 most enriched GO terms (including BP, CC, and MF) and KEGG pathways of DEGs are shown in comparison between groups, **(A, D)** M0 vs M1, **(B, E)** M0 vs M2, and **(C, F)** M1 vs M2. Blue and green in **(A, B, and C)** represent BP and MF, respectively. Data obtained from 3 different donors.

MDMs with Z-stacks, which confirmed bead internalization by MDMs (S1A-S1C Video). These experiments show that *in vitro* differentiation of macrophages also leads to functional differences between these cells.

## 4. Discussion

Dogs have become an increasingly important translational animal model for chronic, inflammatory and immunological disorders [1, 26]. Macrophages play a crucial role in most immune-associated diseases. In countless studies, macrophages derived from monocytes or monocyte-like cell lines are utilized to understand the biology and differentiation of

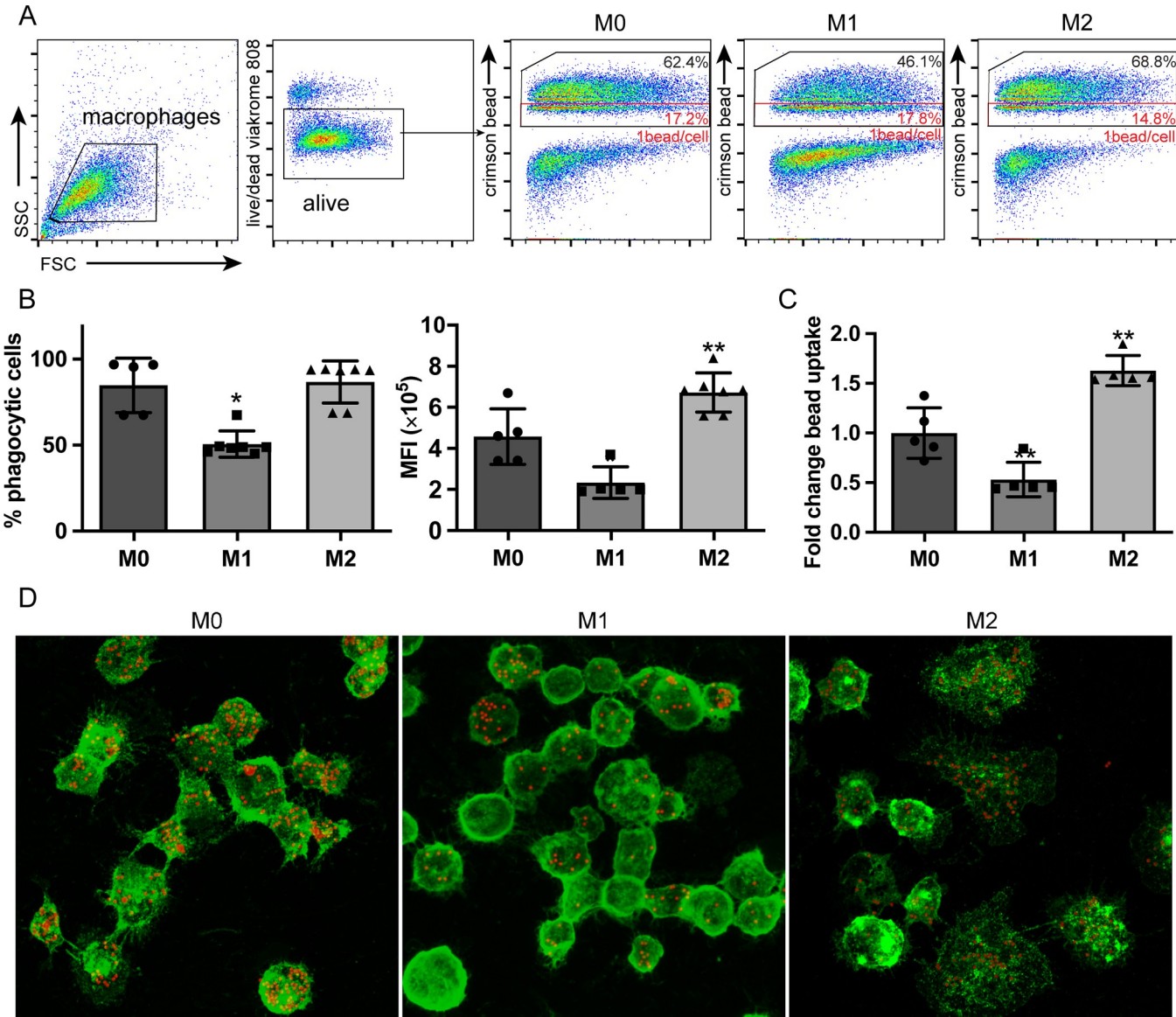

**Fig 6. Phagocytic capability of canine monocyte derived macrophages.** CD14+ monocytes were polarized and activated for 7 days and incubated with crimson carboxylate-modified FluoSpheres fluorescent beads for 4 h. Undifferentiated monocytes (M0) and M1 and M2-polarized MDMs were harvested. (A) MDMs were gated based on their scatter profile (FSC/SSC) and viability (viakrome). The bead content was analyzed quantitatively by flow cytometry for MDMs. MDMs containing beads were gated as indicated by the black box. Cells with 1 bead per cell are indicated by the red box. A representative flow cytometry plot is shown. (B) MDMs containing beads were depicted as the percentage of phagocytic cells and median fluorescence intensity (MFI). (C) The fold change in bead uptake of M1- and M2-polarized MDMs was calculated as described in materials and methods and compared to undifferentiated M0 monocytes. (D) Confocal microscopy was performed to confirm non-specific phagocytosis by MDMs. Cell membrane was stained by WGA-Alexa Fluor 488 shown in green and FluoSpheres fluorescent beads are shown in red. A 3D model of internalization by MDMs was reconstructed from z-stacks. The corresponding video showing 3D structure can be found in S1A-S1C Video. Data obtained from three independent donors measured in triplicates and shown as mean ± SD. Scale bar = 10 μm.

macrophages. So far, most studies have focused on macrophages from humans and mice, and knowledge on canine macrophages is still scarce. Therefore, availability of well-characterized canine macrophages and macrophage subsets is urgently needed for a better understanding of their biological functions. Here, we present a systematic study of canine macrophages derived from monocytes polarized into 2 different states that resemble M1 and M2 macrophages.

Morphology, surface markers, differentially expressed genes and partially functional activity of these macrophage subsets are extensively described, thus providing highly characterized tools for future immunological *in vitro* studies.

The M1 cells derived from monocytes dominantly displayed a round and amoeboid shape, as visualized by light microscopy. Among M2 cells, populations with small round morphology and with elongated spindle shaped morphology exist. These findings parallel the observations in other species such as humans, mouse and pig [27–29]. Also with respect to surface markers, both M1 and M2 cells have many characteristics similar to their human/mouse counterparts. Pan-macrophage markers CD14 and CD11b [30] were highly expressed indicating that these cells indeed maintain macrophage properties during polarization [31]. Co-stimulatory molecules CD40, CD80 and CD86 together with MHC II are essential for antigen presentation and T cell interaction by macrophages and are considered as M1 macrophage markers in many species [31–34]. MHC class II showed increased expression in both M1 and M2 polarized cells. CD163 (scavenger receptor), CD206 (mannose receptor), CD209 (DC-SIGN) were indicated to be specific for M2 macrophages in both mouse and human studies [35–37]. These features were indeed reflected in M2 polarized canine MDMs. In addition, we found that canine M2 macrophages expressed higher levels of CD32, CD64 and CD83 than M1. So far, the discussion on differences in CD32 and CD64 expression levels between M1 and M2 macrophages varies. A variety of studies described CD32 and CD64 as a M1 macrophage marker [38, 39]. Other studies showed that these are highly induced in M2 macrophages [40, 41], or reported that the expression of CD32 by human macrophages did not change after polarization [24].

Next, we compared the overall gene expression profiles of undifferentiated M0 monocytes and polarized M1 and M2 macrophages by RNAseq, using the CD14$^+$ and CD14$^+$ depleted populations as control. M1 cell signature genes include IL-6, IL-1β, TNF-α, iNOS, IL-12 having pro-inflammatory role, while TGF- β, arginase1 and IL-10 are regarded as M2 signature genes playing anti-inflammatory role [10, 13]. These genes were found in canine M1 and M2 cells as well, respectively. The GO and KEGG analyses revealed that many immune related pathways were affected by the polarizing culture conditions. For example when specifically upregulated genes were considered in M1 compared to M0 control cells, significantly enriched GO terms included immune system processes, cytokine and chemokine activity, antigen processing and presentation, G−protein coupled receptor binding, proteasome complexes, while signaling pathways included Epstein−Barr virus infection, tuberculosis, TNF signaling pathway, Th1 and Th2 cell differentiation, the TNF signaling pathway, Th17 cell differentiation, and the chemokine signaling pathway. M1 vs M2 comparison revealed partially similar processes and pathways highlighting that indeed many immune functions were affected by polarization. Although functional specifics cannot yet be deduced, the overwhelming data set does provide a detailed database for prediction and explanation of macrophage subset behavior in experimental immunological settings.

The change of phenotypes and gene profiles is often accompanied by a change of functional activity [42], and ultimately the functionality of immune cells is arguably the most important readout for immunological characterization. Phagocytosis is one of the main functions of macrophages, which is modulated by a variety of phagocytic receptors. Using latex beads, a change in phagocytosis capacity was observed with M2-polarized MDMs exhibiting the strongest phagocytic capacity and M1 having reduced phagocytic capacity compared to M0 monocytes. In agreement with these results, other studies have also shown that human M2 macrophages displayed a higher phagocytosis capacity than M1 macrophages [24, 43, 44]. However, in contrast with our results, other studies have demonstrated that human M1 macrophages actually showed a higher level of phagocytic activity [45–47].

The methodology for polarization of macrophages using GM-CSF/IFN-γ/LPS and M-CSF/IL-4 is widely used in literature. The contribution of each of these factors to polarization is not well described for canine macrophages, but the growth factors GM-CSF and M-CSF likely have a prominent role in this. Macrophages are actually unique in responding to 2 different growth factors [48], where M-CSF (also known as CSF-1) binds its receptor CD155 activating several pathways including ERK, PI3K, PKC and MAPK. GM-CSF (CSF-2) signals through a different receptor, CD116 activating STAT5, ERK, NF-B, and IRF5 [17]. Indeed in human studies monocyte to macrophage maturation in the presence of M-SCF or GM-CSF only already led to large phenotypic differences [49, 50] and they are in itself already considered M1 and M2 stimuli [17]. However, in transcriptome analyses, addition of IL-4 to M-CSF stimulated macrophages led to an additional upregulation of 104 transcripts overly representing M2-associated cytokines [51]. For GM-CSF alone stimulate macrophages many M1 genes were upregulated, including typical M1 cytokines, but addition of LPS (signaling through TLR4) heavily increased the production of several typical M1 cytokines such as IL6 and IL-1β [17]. Although there is no consensus for the perfect polarization method, our experimental setup was specifically chosen to enable a good comparison of characteristics between canine polarized macrophages and macrophages of other species described in literature. In addition an important goal of these studies was also to determine the robustness and reproducibility of producing canine monocyte derived M1 and M2 macrophages. Therefore our methodology to obtain monocyte derived macrophages was largely similar in setup to earlier studies [18, 19] with the exception that our study included CD14[+] cells and, compared to Chow et al., used different growth factors for polarization. Nevertheless, overall, obtained M1 and M2 subsets were comparable between all studies with respect to morphology, and upregulation of several M1 and M2 surface markers. For example, expression of CD206 was consistently higher in M2 than M1 macrophages in all studies. However, there were also several differences, such as in MHCII and CD40 expression levels on M1, and in CD32 and CD163 levels on M2 macrophages. At gene expression level, the fold-differences in gene expression (using microRNA) were particularly high in the study of Heinrich et al. [18], but numbers of genes affected were comparable. Remarkably, a comparison of Gene Ontology and KEGG-pathways between studies shows that not one or two specific pathways stand out that define M1 or M2 macrophages. Also completely opposite outcomes were observed regarding the phagocytic capacity of M1 compared to M2 macrophages between the current study and Chow et al. [19] These differences probably reflect the plasticity of macrophages and perhaps small differences in experimental setup. In addition, genetic differences between donors and their health status may contribute to the observed differences. This observed variability indicates that one should be very careful in employing specific surface or gene expression markers as polarization markers, based on a limited number of studies, and that reproducibility of macrophage polarization is actually quite complex. An additional feature, not often tested, is whether the polarized macrophages have retained their plasticity. In other words, M1 macrophages should be still be able to revert to M2 or any other macrophage subtype if the external milieu is changed [52]. This adds another layer of complexity to *in vitro* culture of macrophages with *in vivo* characteristics.

There is a growing general interest in studying- or using polarized canine macrophages and this is reflected in the literature. For example, canine macrophages can be used as infection models in which polarization can play an important role. *Leishmania* is an important pathogen of dogs and both the DH82 cell line [53] as well as canine MDMs [54] were used to study the effect of *Leishmania* infection. Canine MDMs were also used to study *Mycobacterium intracellulare* infection, which polarized the MDM towards M1 [55]. Besides microorganisms changing polarization of macrophages, also communication between immune cells leading to polarization has been studied in which canine B-cells were able to switch M1 macrophages to

M2 -like macrophages [56]. Finally the effect of extracellular vesicles on canine macrophages [57] or canine M1 macrophage EVs themselves on apoptosis [58] have also been described. These examples indicate again that there is a need for a well-defined and well described macrophage model system to obtain reliable and well-interpretable results.

In conclusion, in this study we successfully developed a polarization protocol for canine monocytes, and the resulting M1 and M2 macrophage types were comprehensively characterized. The two subsets differ significantly in morphology, gene and protein expression, and functionality and are an excellent model for future research on (polarized) canine macrophage studies.

## Supporting information

**S1 Table. Gene counts of M0 monocytes and M1, M2 macrophages and CD14$^+$ and CD14D cells.**
(XLSX)

**S2 Table. Differentially expressed genes of M0 monocytes, M1, M2 macrophages and CD14$^+$ and CD14D cells.**
(XLSX)

**S3 Table. Top 50 DEG pairwise comparisons M0, M1, M2 and CD14$^+$ cells.**
(XLSX)

**S1 Fig. Identification of non-viable cells by back gating.**
(PDF)

**S2 Fig. Selection of Immunity-related DEG's.**
(TIF)

**S1 Video.** A. Bead uptake M0 monocytes visualized by confocal microscopy. B. Bead uptake M1 MDMs visualized by confocal microscopy. C. Bead uptake M2 MDMs visualized by confocal microscopy.
(ZIP)

## Acknowledgments

The authors thank Evieke Ruijsink for collecting flow cytometry data from one of the donor dogs. All fluorescent microscopy images have been acquired at the Center of Cellular Imaging, Faculty of Veterinary Medicine, Utrecht University. All the flow cytometry data were collected using the Flow Cytometry and Cell Sorting Facility at the department of Infectious Diseases & Immunology in Utrecht University.

## Author Contributions

**Conceptualization:** Willem van Eden, Alice J. A. M. Sijts, Femke Broere.

**Data curation:** Irene S. Ludwig.

**Formal analysis:** Femke Broere.

**Methodology:** Qingkang Lyu.

**Supervision:** Edwin J. A. Veldhuizen, Victor P. M. G. Rutten, Alice J. A. M. Sijts, Femke Broere.

**Validation:** Edwin J. A. Veldhuizen.

**Writing – original draft:** Qingkang Lyu.

**Writing – review & editing:** Edwin J. A. Veldhuizen, Victor P. M. G. Rutten, Alice J. A. M. Sijts, Femke Broere.

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
