## [Decision Letter · Decision Letter 0]

28 Feb 2023

PONE-D-23-02588Characterization of polarization states of canine monocyte derived macrophages.PLOS ONE

Dear Dr. Veldhuizen,

Thank you for submitting your manuscript to PLOS ONE. After careful consideration, we feel that it has merit but does not fully meet PLOS ONE’s publication criteria as it currently stands. Therefore, we invite you to submit a revised version of the manuscript that addresses the points raised during the review process.

We look forward to receiving your revised manuscript.

Kind regards,

Kenji Fujiwara, PhD, MD

Academic Editor

PLOS ONE

Journal Requirements:

**Additional Editor Comments:**

Dear Dr. Veldhuizen.

The article is about the characteristics of canine macrophages. The results seem almost similar to human macrophages, and the similarity could attract many readers. The manuscript is well-written and I think this is eligible to be considered for major revision by agreeing with the reviewers. I added some minor concerns below.

I look forward to the revised one.

Best regards,

Kenji Fujiwara

Minor concerns

1. American English may be appropriate. For example, “RNAseq of the three macrophage subsets showed distinct gene expression profiles, which are closely associated with immune responsiveness, cell differentiation and phagocytosis.”

2. In the abstract, “M2- polarized MDMs” suddenly appeared. Please show elaboration of MDM at first appearance.

3. I think PLOS ONE does not ask to add keywords on the title page. Also, figure legends and a table should be in the result section. Please check the guideline.

https://journals.plos.org/plosone/s/submission-guidelines

4. In Figure 3A, the numbers in the Venn diagrams seem too small. Could you change them to larger for readers? This issue is the same for Figures 4 and 5.

Reviewers' comments:

Reviewer's Responses to Questions

**Comments to the Author**

1. Is the manuscript technically sound, and do the data support the conclusions?

Reviewer #1: Yes

Reviewer #2: Partly

2. Has the statistical analysis been performed appropriately and rigorously? 

Reviewer #1: Yes

Reviewer #2: Yes

3. Have the authors made all data underlying the findings in their manuscript fully available?

Reviewer #1: Yes

Reviewer #2: Yes

4. Is the manuscript presented in an intelligible fashion and written in standard English?

Reviewer #1: Yes

Reviewer #2: Yes

5. Review Comments to the Author

Reviewer #1: 

The manuscript "Characterization of polarization states of canine monocyte-derived macrophages" is well-written and experimentally well-conducted.

Canine macrophage polarization still has a poor level of scientific understanding. In this study, the authors polarized canine monocytes toward M1 and M2 macrophages and contrasted them to unpolarized M0 cells. The authors characterized polarized M1 and M2 and unpolarized M0 macrophages by morphology, gene profiles, surface marker features, and functional properties. The following analysis revealed that canine macrophages M1, M2, and M0 differ from one another and can be identified based on their phenotypic characteristics and the expression of particular markers, just like macrophages from other well-studied species. RNAseq revealed the three macrophage subsets' gene expression patterns closely related to phagocytosis, cell differentiation, and immune responsiveness. No new polarization markers could, however, be assigned by the authors.

Future research on (polarized) canine macrophage studies will benefit significantly from monocyte-derived polarized macrophages, particularly in studies of immune-related diseases in dogs. The differentiation protocols produced two subsets of macrophages (M1 and M2) that differ significantly in morphology, gene and protein expression, and functionality. These subsets are also a great model for future research on macrophage studies in general.

Recommendations.

The manuscript is accepted with minor revisions.

1) Discussion: add more biological information about DEGs found in M1 and M2 macrophages.

2) Add more information regarding the significance of the DEGs specific for each subset of macrophages.

Reviewer #2: Dear,

The authors explored an interesting research field and the main goal of the manuscript is really necessary to understand the macrophage polarization for canine monocytes. The authors proposed a protocol to characterize the macrophage polarization, and the main approaches were considered but it should be improved. The major concern is the basic experimental design to induce the macrophage polarization because the authors called M0 macrophages a cell type that was not differentiated to macrophage and is a monocyte. Moreover, the authors should demonstrated that most of the results found were not already induced by M/GM-CSF, then control groups containing only M-CSF or GM-CSF is strongly recommended. LPS can also induce some characteristics related to the M2 macrophages for murine and human cells in a dose-dependent manner, then it should be discussed by the authors or demonstrate that 100ng/mL of LPS is adequate to evaluate only M1 macrophage. The association of IL-12 and IFN-g could be a better option for M1 macrophage induction. So, the impact of LPS/IFN and IL-4 in the macrophage polarization is confused with M/GM-CSF and so far there is no M0 macrophage.

In addition, iNOs and Arginase-1 were missed and must be evaluated to complement the flow cytometry data, and both mRNA relative expression and protein expression are well accepted. The morphological analysis should be improved, and H&E staining is an example for that. The phagocytosis assay could consider the incubation with IFN-g alone for each condition as a positive control to facilitate the engulfment of beads.

The plasticity of canine monocytes should be validated by the repolarization assay, and the balance between iNOS/Arginase-1 is a great parameter to demonstrate that M1/M2 subsets can be repolarized.

6. PLOS authors have the option to publish the peer review history of their article (what does this mean?). If published, this will include your full peer review and any attached files.

Reviewer #1: **Yes: **MARCO ANTONIO MERAZ RIOS

Reviewer #2: No

---

## [Author Response · Author response to Decision Letter 0]

6 Jul 2023

editorial remarks

Minor concerns

1. American English may be appropriate. For example, “RNAseq of the three macrophage subsets showed distinct gene expression profiles, which are closely associated with immune responsiveness, cell differentiation and phagocytosis.”

Reply: We corrected the text to American English where appropriate. 

2. In the abstract, “M2- polarized MDMs” suddenly appeared. Please show elaboration of MDM at first appearance.

Reply: We removed MDMs from the abstract. In the manuscript itself it is now described as monocyte-derived macrophages upon its first appearance Line 116.

3. I think PLOS ONE does not ask to add keywords on the title page. Also, figure legends and a table should be in the result section. Please check the guideline.

https://journals.plos.org/plosone/s/submission-guidelines

Reply: Keywords have been removed and the figure legends and table have been moved to the correct position in the manuscript text. 

4. In Figure 3A, the numbers in the Venn diagrams seem too small. Could you change them to larger for readers? This issue is the same for Figures 4 and 5.

Reply: We have changed the fonts in Fig 3- 5 to make them easier to read. 

Reviewer #1: 

The manuscript "Characterization of polarization states of canine monocyte-derived macrophages" is well-written and experimentally well-conducted.

Canine macrophage polarization still has a poor level of scientific understanding. In this study, the authors polarized canine monocytes toward M1 and M2 macrophages and contrasted them to unpolarized M0 cells. The authors characterized polarized M1 and M2 and unpolarized M0 macrophages by morphology, gene profiles, surface marker features, and functional properties. The following analysis revealed that canine macrophages M1, M2, and M0 differ from one another and can be identified based on their phenotypic characteristics and the expression of particular markers, just like macrophages from other well-studied species. RNAseq revealed the three macrophage subsets' gene expression patterns closely related to phagocytosis, cell differentiation, and immune responsiveness. No new polarization markers could, however, be assigned by the authors.

Future research on (polarized) canine macrophage studies will benefit significantly from monocyte-derived polarized macrophages, particularly in studies of immune-related diseases in dogs. The differentiation protocols produced two subsets of macrophages (M1 and M2) that differ significantly in morphology, gene and protein expression, and functionality. These subsets are also a great model for future research on macrophage studies in general.

Recommendations.

The manuscript is accepted with minor revisions.

1) Discussion: add more biological information about DEGs found in M1 and M2 macrophages.

Reply: We have added several lines on the biological information in the discussion. Line 481-484.

2) Add more information regarding the significance of the DEGs specific for each subset of macrophages.

Reply: From the RNAseq data we selected gene expression levels of some described M1 or M2 markers. This shows that indeed, for example Il-6 and IL-1β are much higher in M1 compared to M2 while TGF- β is higher in M2. Surprisingly arginase expression does not show major differences. We added this as a supplementary figure to the manuscript and describe the results in lines 358-363.

Reviewer #2: Dear,

1) The authors explored an interesting research field and the main goal of the manuscript is really necessary to understand the macrophage polarization for canine monocytes. The authors proposed a protocol to characterize the macrophage polarization, and the main approaches were considered but it should be improved. The major concern is the basic experimental design to induce the macrophage polarization because the authors called M0 macrophages a cell type that was not differentiated to macrophage and is a monocyte.

Reply: We thanks the reviewer for the nice remarks and indeed agree that M0 are not macrophages. We have now changed the terminology throughout the whole manuscript to either M0 monocytes or M0 cells.

However, regarding the 1st remark about the basic design, (and this will come back in the rebuttal below), we feel that our aim of the manuscript was not clearly brought across. We do not seek to optimize the protocol to make polarized canine macrophages, but instead we want to characterize the polarized macrophages that are obtained using a standardized method. This method is well described in human and mouse, and recently a very small number of canine studies also used this protocol based on GM-CSF/IFNγ/LPS and M-CSF/IL-4. Finding the best protocol is a completely different study requiring a completely different set-up. In addition, we would not know how to clearly define ‘an improved method’ because it would be based on human and mouse macrophage readouts and would deny possible species differences in macrophage polarization and functionality. We added a few sentences in the discussion to emphasize this aim of our study again. Line 506-511.

2) Moreover, the authors should demonstrated that most of the results found were not already induced by M/GM-CSF, then control groups containing only M-CSF or GM-CSF is strongly recommended. 

Reply: We performed these experiments as requested and polarized M0 cells towards M1/M2 using either GM-CSF / M-CSF alone, or with the addition IFN-γ/LPS and IL-4. We performed qPCR on specific M1 and M2 markers and also checked the cell’s morphology. These experiments showed that indeed the presence of the growth factors alone already had strong polarizing activity, as seen by the morphology, see Figure E(xtra)1 below . However, the ‘extra’ addition of IFN-γ / LPS resulted in an upregulation of IL-6 (M1 marker) compared to GM-CSF alone, while addition of IL-4 induced gene expression of TGF-beta and MS4A2 (M2 marker), showing that these factors have an additional effect on gene expression of polarization marker genes (Fig. E2). These actual experiments are, however outside the scope of the manuscript. We have included a comment on our findings in the discussion. Line 532-536.

(see figure in 'response to reviewer file at end of document'; figures couldn't be copied in this section)

Fig E1. Morphology of M1 and M2 macrophages on day 8 after polarization. Polarization with GM-CSF with or without IFNγ/LPS resulted in a M1 phenotype. Polarization with M-CSF with or without IL-4 resulted in a M2 phenotype. 

(see figure in 'response to reviewer file at end of document'; figures couldn't be copied in this section)

 Fig. E2. Relative gene expression of cells matured in the presence of A) GM-CSF ± IFN/LPS and B) M-CSF ± IL-4. 

3) LPS can also induce some characteristics related to the M2 macrophages for murine and human cells in a dose-dependent manner, then it should be discussed by the authors or demonstrate that 100ng/mL of LPS is adequate to evaluate only M1 macrophage. 

Reply: We agree with the reviewer that LPS indeed can cause some M2 characteristics. Before performing our study on monocyte derived macrophages we did some initial studies on the effect on LPS on canine macrophages in the macrophage cell line 030D which indicated that this concentration indeed induced classical M1 markers. Eventually, the results of our monocyte derived macrophages polarization also shows that this methodology indeed results in M1 like features.

4) The association of IL-12 and IFN-g could be a better option for M1 macrophage induction. So, the impact of LPS/IFN and IL-4 in the macrophage polarization is confused with M/GM-CSF and so far there is no M0 macrophage.

Reply: We agree with the reviewer that several variables in the polarization methodology have not been explored, including the use of IL-12 and IFNy. However, as stated above, the focus of this study is not to optimize the polarization method.

We agree with the wrong use of the term M0 macrophages and this has been replaced throughout the manuscript.

5) In addition, iNOs and Arginase-1 were missed and must be evaluated to complement the flow cytometry data, and both mRNA relative expression and protein expression are well accepted. 

Reply: We have added the gene expression of several key genes in supplementary Figure S1 , including iNOS and Arginase1 and discuss these in the manuscript, line 358-363. Unfortunately we don’t have assays running for determination of these enzymes on a protein level (much less tools are available for canine research than mouse and human). 

6) The morphological analysis should be improved, and H&E staining is an example for that. 

Reply: Unfortunately , we made an error in the morphological analyses of our polarized macrophages in our new set of experiments, and only stained with Hematoxylin. We added these stainings below for your convenience but realize they don’t add any value. However, we are not completely sure what the reviewer would like to see with a correct H&E staining. If it is essential, we are willing to obtain fresh canine blood, polarize monocytes to macrophages and perform the analysis. But considering the infrequency of obtaining fresh buffy coats from (canine) patients we doubt if the extra effort is really required. 

7) The phagocytosis assay could consider the incubation with IFN-g alone for each condition as a positive control to facilitate the engulfment of beads.

Reply: We agree that many variables can be tested to optimize phagocytosis or other characteristics of polarized macrophages. But this falls out of the scope of the current manuscript which is a characterization of in vitro M1 and M2 polarized macrophages in dogs. The contribution of each polarization-inducing compound is subject to a follow up study. 

8) The plasticity of canine monocytes should be validated by the repolarization assay, and the balance between iNOS/Arginase-1 is a great parameter to demonstrate that M1/M2 subsets can be repolarized.

Reply: In our extra set of experiments of experiments we used our day8 M1 and M2 polarized macrophages and treated them with the opposite polarizing environments. So M2 polarized macrophages were grown for an additional 2 days in the presence of GM-CSF/ IFN-γ and LPS and day 8 M1 polarized macrophages were grown an additional 2 days in the presence of M-CSF and Il-4. Control M1 and M2 macrophages were grown an additional 2 days in their original growth medium. Clear morphological changes were observed for M2� M1 repolarization (Fig. E3), and also gene expression of IL-6, was upregulated (Fig. E4). The M1�M2 repolarization was more subtle in both aspects. Morphological changes were less clear, and we didn’t see the typical M2-like morphology. However, on a gene level a tendency towards more MS4A2 (M2 marker) gene expression was observed. Based on these results we can conclude that at day 8 both M1 and M2 had not reached an end-point polarization state and still had plasticity. However, we cannot state that they repolarized to classical M1 or M2 macrophages. We made a comment about the importance of repolarization in the discussion in Line 532-536. We will pursue the re-polarization study in our planned follow up studies, also optimizing this repolarization protocol. 

(see figure in 'response to reviewer file at end of document'; figures couldn't be copied in this section) 

Fig. E3. Morphology of repolarized macrophages on day 10 after (re-)polarization. Day 8 M1 macrophages did not clearly change morphology after 2 days repolarization with M2 conditions (top left vs top right image). M2 macrophages on the other hand clearly depicted a mor eM1-like morphology after repolarization in M1 conditions (compare bottom left and bottom right image. Fig. E4. Relative gene expression of cells matured for 8 days towards M1 or M2, and on day 9 repolarized towards opposite polarization state. ________________________________________

6. PLOS authors have the option to publish the peer review history of their article (what does this mean?). If published, this will include your full peer review and any attached files.

---

## [Decision Letter · Decision Letter 1]

31 Jul 2023

PONE-D-23-02588R1Characterization of polarization states of canine monocyte derived macrophages.PLOS ONE

Dear Dr. Veldhuizen,

Thank you for submitting your manuscript to PLOS ONE. After careful consideration, we feel that it has merit but does not fully meet PLOS ONE’s publication criteria as it currently stands. Therefore, we invite you to submit a revised version of the manuscript that addresses the points raised during the review process.

I think the manuscript was revised well. One reviewer asked some questions and I agree with the concerns. After receiving minor revisions, we will reevaluate again.

We look forward to receiving your revised manuscript.

Kind regards,

Kenji Fujiwara, PhD, MD

Academic Editor

PLOS ONE

Journal Requirements:

Additional Editor Comments:

Dear Dr. Veldhuizen.

I think the manuscript was revised well. One reviewer asked some questions and I agree with the concerns. After receiving minor revisions, we will reevaluate again.

Best,

Kenji Fujiwara

Reviewers' comments:

Reviewer's Responses to Questions

**Comments to the Author**

1. If the authors have adequately addressed your comments raised in a previous round of review and you feel that this manuscript is now acceptable for publication, you may indicate that here to bypass the “Comments to the Author” section, enter your conflict of interest statement in the “Confidential to Editor” section, and submit your "Accept" recommendation.

Reviewer #1: All comments have been addressed

Reviewer #3: (No Response)

2. Is the manuscript technically sound, and do the data support the conclusions?

Reviewer #1: Yes

Reviewer #3: Yes

3. Has the statistical analysis been performed appropriately and rigorously? 

Reviewer #1: Yes

Reviewer #3: Yes

4. Have the authors made all data underlying the findings in their manuscript fully available?

Reviewer #1: Yes

Reviewer #3: Yes

5. Is the manuscript presented in an intelligible fashion and written in standard English?

Reviewer #1: Yes

Reviewer #3: Yes

6. Review Comments to the Author

Reviewer #1: The reviewers' criticisms have been appropriately resolved by the authors, and the work is now suitable for publishing.

Reviewer #3: The manuscript "Characterization of polarization states of canine monocyte-derived macrophages" provides phenotypic descriptions of polarization of canine macrophages in pro-inflammatory M1 and pro-resolution M2 states. Authors used well characterized methods to induce their polarization, allowing for more direct comparison between their canine model and other mammal models such as human and mouse. Authors also measured well characterized endpoints including phagocytosis, cell surface receptor levels, and RNAseq plus GO/KEGG analysis which provides strong evidence for similarities between canine macrophage polarization states and human/mouse macrophages. The findings described within are interesting and overall, well presented, however the use of M0 to describe undifferentiated monocytes remains an issue. While authors have modified their description in the methods and in most instances in the rest of the text, some instances remain unchanged from the original description. Regardless, I believe the use of M0 to describe undifferentiated monocytes introduces unnecessary confusion due to its usual use to describe unpolarized macrophages following differentiation from monocytes and should be removed entirely, replaced with “undifferentiated monocytes (UM)” or similar. I would also recommend additional citations, particularly in the discussion, when referencing the use of these methods in other species as well as known or suspected responses in other species.

Major comments:

1. Based on the methods description and to avoid confusion with terminology used in the literature, the use of M0 for undifferentiated monocytes should be removed and replaced with “undifferentiated monocytes (UM)” or similar. If authors insist on maintaining this terminology, careful attention should be paid to their use of M0, as its use in numerous places incorrectly stated or (to me) implied they were in fact monocyte-derived macrophages. See line 31, 279, 305, 417-419, 479. Further, in the abstract and throughout “un differentiated” should be changed to “undifferentiated”. Each figure legend should also clearly indicate that M0=undifferentiated monocyte.

2. I disagree with the author’s reply to reviewer 2, comment 2 in regard to excluding the results of their additional experiment from the manuscript. GM-CSF/M-CSF alone inducing M1/M2-like polarization absent additional stimuli has been previously described in other models and appears true for authors’ model as well. Their description of IL-6/TGF-b/MS4A2 upregulation following additional stimuli suggests the addition of IFNy+LPS/IL4 is adding to the polarization as in other models. If the authors do not feel these specific results are appropriate to include, I would recommend they at least add a comment to the discussion (with appropriate citations) to describe this known phenomenon. I believe this was attempted in lines 506-511, but citations should be included and comments describing findings in human/mouse cells should be used to compare to these findings, despite possible differences between responses due to species.

3. While videos were referenced throughout the manuscript, especially in Figure 6 legend, they are not listed under “Supplementary Files” and were not linked to the review. I was unable to access supplemental materials Video S1A, B, and C and therefore cannot comment on them.

4. For transparency, please add individual dots for each subject onto new Figure S1 (currently bar plot, but would be ideally shown with mean, error, and individual data points). This should also be applied to bar plots in figures 1C, 2B and 6B. Currently unclear whether N=3 was averaged across triplicate or whether all 9 datapoints were used to create graphs, the former is the most appropriate.

Minor Comments:

5. Please provide more information on the donor dogs in section 2.1. A clear listing of how many samples were collected, age, the sex of the animal, and optionally the breed should be listed.

6. Line 190 states “Transcripts with read counts <2 were filtered out for DEG testing.” Please clarify if this is filtering based on single sample (any sample <2 excludes gene for all samples), all samples (all samples had to be >=2 or excludes gene), or mean counts (assessed by group or of all samples).

7. In Figure 1B, the M1 macrophage panel shows a sizable population of events at approximately 35 degrees which is not present in M0 or M2 panels. This appears independent of what I assume is debris which is present in all panels including M1 at approximately 20 degrees. Do the authors have any information about this population, i.e. did they attempt to measure live/dead of this population to determine if these are dead/dying cells? This relates to Figure 6A which uses a representative panel to describe gating which does not possess this population.

8. Replace “Representable” with “Representative” on line 307.

9. Figure 3B may benefit from vertical breaks between samples to allow for easier comparison between replicates. In the legend for 3B please change to “differentially expressed genes” and indicate whether p-value was also used as a cutoff for inclusion in the heatmap.

10. Please add that PCA was created using FPKM in Figure 4’s legend.

11. Unsure what authors mean on line 399-402. Consider rephrasing.

12. The last sentence of the conclusion should read “…in morphology, gene and protein expression, and functionality…”

13. In Figure 6A, is there a horizontal gap between cell populations allowing for a 1 bead/cell and multiple beads/cell events that was used to draw gates? I am unable to assess this due to the panel size/resolution.

14. Lines 481-483. I generally agree, please add citations of these previous findings.

15. Line 494. “Deducted” should be “deduced”.

16. Line 532-536. I agree with this statement, but citations should be provided to describe previous findings suggesting this possibility.

17. Line 543-544. This sentence is confusing as is. Please consider rephrasing.

7. PLOS authors have the option to publish the peer review history of their article (what does this mean?). If published, this will include your full peer review and any attached files.

Reviewer #1: **Yes: **MARCO ANTONIO MERAZ RIOS

Reviewer #3: No

---

## [Author Response · Author response to Decision Letter 1]

29 Aug 2023

Major comments:

1. Based on the methods description and to avoid confusion with terminology used in the literature, the use of M0 for undifferentiated monocytes should be removed and replaced with “undifferentiated monocytes (UM)” or similar. If authors insist on maintaining this terminology, careful attention should be paid to their use of M0, as its use in numerous places incorrectly stated or (to me) implied they were in fact monocyte-derived macrophages. See line 31, 279, 305, 417-419, 479. Further, in the abstract and throughout “un differentiated” should be changed to “undifferentiated”. Each figure legend should also clearly indicate that M0=undifferentiated monocyte.

We decided to maintain the M0 terminology to depict the common origin as the M1 and M2 macrophages, but we replaced ‘M0 cells’ with ‘undifferentiated monocytes’ at multiple (>20) places in the manuscript, including all the figure legends. This should clarify even more that M0 cells are indeed monocytes and not macrophages. The spaces between ‘un’ and ‘differentiated’have been removed.

2. I disagree with the author’s reply to reviewer 2, comment 2 in regard to excluding the results of their additional experiment from the manuscript. GM-CSF/M-CSF alone inducing M1/M2-like polarization absent additional stimuli has been previously described in other models and appears true for authors’ model as well. Their description of IL-6/TGF-b/MS4A2 upregulation following additional stimuli suggests the addition of IFNy+LPS/IL4 is adding to the polarization as in other models. If the authors do not feel these specific results are appropriate to include, I would recommend they at least add a comment to the discussion (with appropriate citations) to describe this known phenomenon. I believe this was attempted in lines 506-511, but citations should be included and comments describing findings in human/mouse cells should be used to compare to these findings, despite possible differences between responses due to species.

We don’t feel completely comfortable adding our data in the current form, both because we would include many more samples for firm conclusions (and statistical significance of ALL affected genes), and because of the fact that it actually distracts from the focus of the manuscript which is to compare human vs canine polarized macrophages (instead of deciphering and optimizing canine macrophage polarization itself. However, we added a paragraph in the discussion in which we describe this phenomenon known in human macrophage polarization, thus large effects of M-CSF and GM-CSF alone, but additional polarizing effects upon addition of LPS or IL-4, including the appropriate references. Lines 517-529 of track-changes version of the manuscript. 

3. While videos were referenced throughout the manuscript, especially in Figure 6 legend, they are not listed under “Supplementary Files” and were not linked to the review. I was unable to access supplemental materials Video S1A, B, and C and therefore cannot comment on them.

Apparently something went wrong during uploading of the videos . They are now included.

4. For transparency, please add individual dots for each subject onto new Figure S1 (currently bar plot, but would be ideally shown with mean, error, and individual data points). This should also be applied to bar plots in figures 1C, 2B and 6B. Currently unclear whether N=3 was averaged across triplicate or whether all 9 datapoints were used to create graphs, the former is the most appropriate.

We agree with the reviewer and have added the individual data points to figures 1, 2 and 6 as suggested. Regarding supplementary figure S1:, we picked M1 and M2 signature genes from bulk RNA-seq from 3 donors. We then used R package DESeq2 for differential analysis. In this package, experimental repetition was considered already but the outcome is only one value of log2 fold change. Hence this figure has no error bar. 

Minor Comments:

5. Please provide more information on the donor dogs in section 2.1. A clear listing of how many samples were collected, age, the sex of the animal, and optionally the breed should be listed.

We added this information to the materials section. Line 100-102 (of track-changes)

6. Line 190 states “Transcripts with read counts <2 were filtered out for DEG testing.” Please clarify if this is filtering based on single sample (any sample <2 excludes gene for all samples), all samples (all samples had to be >=2 or excludes gene), or mean counts (assessed by group or of all samples).

If one gene has read counts lower than 2 fold difference in all samples, this gene was filtered out. We clarified this now in the Materials section LINE 192 (of track changes)

7. In Figure 1B, the M1 macrophage panel shows a sizable population of events at approximately 35 degrees which is not present in M0 or M2 panels. This appears independent of what I assume is debris which is present in all panels including M1 at approximately 20 degrees. Do the authors have any information about this population, i.e. did they attempt to measure live/dead of this population to determine if these are dead/dying cells? This relates to Figure 6A which uses a representative panel to describe gating which does not possess this population.

We are indeed pretty sure that population represents dead cells, so we exclude them at the beginning.. Please see the back-gating data below.

8. Replace “Representable” with “Representative” on line 307.

We made the appropriate change (which we thought was quite a funny mistake)

9. Figure 3B may benefit from vertical breaks between samples to allow for easier comparison between replicates. In the legend for 3B please change to “differentially expressed genes” and indicate whether p-value was also used as a cutoff for inclusion in the heatmap.

We agree with the suggestion and added a vertical line among groups in figure 3. We made the textual change to differentially expressed genes, LINE 344 (track changes). In our analysis, the adjusted p value < 0.05 was used as a cutoff. We added an appropriate statement in the Figure 3B legend. LINE 344-345 (track changes)

10. Please add that PCA was created using FPKM in Figure 4’s legend.

We added this to the figure legend LINE 385 (TT)

11. Unsure what authors mean on line 399-402. Consider rephrasing.

We rephrased the sentence to clarify that there is no clear result in this analysis that indicates major immunological differences between M0 and M2. LINE 403-406 (TT)

12. The last sentence of the conclusion should read “…in morphology, gene and protein expression, and functionality…”

We made the suggested change LINE 572-573 (TT)

13. In Figure 6A, is there a horizontal gap between cell populations allowing for a 1 bead/cell and multiple beads/cell events that was used to draw gates? I am unable to assess this due to the panel size/resolution.

When the figures are ‘blown up’you can indeed see a gap between 1 bead and multiple beads, see figure below. We indicated this in the figure and the legend but have not commented on the 1 bead – multiple bead ratio (which was similar) since we think % positive cells and total MFI is more important. 

14. Lines 481-483. I generally agree, please add citations of these previous findings.

we added 2 references to this statement. LINE 491 (TT)

15. Line 494. “Deducted” should be “deduced”.

we made the suggested change LINE 502 (TT)

16. Line 532-536. I agree with this statement, but citations should be provided to describe previous findings suggesting this possibility.

The reference describing repolarization has been added LINE 555 (TT)

17. Line 543-544. This sentence is confusing as is. Please consider rephrasing.

We have rephrased the sentence to clarify the statement. LINE 563-564(TT)

---

## [Decision Letter · Decision Letter 2]

21 Sep 2023

PONE-D-23-02588R2Characterization of polarization states of canine monocyte derived macrophages.PLOS ONE

Dear Dr. Veldhuizen,

Thank you for submitting your manuscript to PLOS ONE. After careful consideration, we feel that it has merit but does not fully meet PLOS ONE’s publication criteria as it currently stands. Therefore, we invite you to submit a revised version of the manuscript that addresses the points raised during the review process.

We look forward to receiving your revised manuscript.

Kind regards,

Kenji Fujiwara, PhD, MD

Academic Editor

PLOS ONE

Journal Requirements:

Additional Editor Comments (if provided):

Dear Dr. Veldhuizen

The article was reviewed by the previous reviewer and minor revision was recommended. I agree with it. I look forward to the updated one.

Best regards,

Kenji Fujiwara

Academic editor

Reviewers' comments:

Reviewer's Responses to Questions

**Comments to the Author**

1. If the authors have adequately addressed your comments raised in a previous round of review and you feel that this manuscript is now acceptable for publication, you may indicate that here to bypass the “Comments to the Author” section, enter your conflict of interest statement in the “Confidential to Editor” section, and submit your "Accept" recommendation.

Reviewer #3: (No Response)

2. Is the manuscript technically sound, and do the data support the conclusions?

Reviewer #3: Yes

3. Has the statistical analysis been performed appropriately and rigorously? 

Reviewer #3: Yes

4. Have the authors made all data underlying the findings in their manuscript fully available?

Reviewer #3: Yes

5. Is the manuscript presented in an intelligible fashion and written in standard English?

Reviewer #3: Yes

6. Review Comments to the Author

Reviewer #3: Overall, the authors have thoroughly addressed the majority of the reviewer comments. However, there are still a few items that require further clarification.

1. Prior comment by reviewer: Line 190 states “Transcripts with read counts <2 were filtered out for DEG testing.” Please clarify if this is filtering based on single sample (any sample <2 excludes gene for all samples), all samples (all samples had to be >=2 or excludes gene), or mean counts (assessed by group or of all samples).

author response: If one gene has read counts lower than 2 fold difference in all samples, this gene was filtered out. We clarified this now in the Materials section LINE 192 (of track changes)

New reviewer comment: It’s the reviewer’s understanding that based on the manuscript changes the authors meant “…read counts lower than 2 in all samples…” rather than “2 fold difference”, please affirm whether this is true. If 2 fold difference was intended, please edit the manuscript accordingly.

2. Prior Comment by reviewer: In Figure 1B, the M1 macrophage panel shows a sizable population of events at approximately 35 degrees which is not present in M0 or M2 panels. This appears independent of what I assume is debris which is present in all panels including M1 at approximately 20 degrees. Do the authors have any information about this population, i.e. did they attempt to measure live/dead of this population to determine if these are dead/dying cells? This relates to Figure 6A which uses a representative panel to describe gating which does not possess this population.

author response: We are indeed pretty sure that population represents dead cells, so we exclude them at the beginning.. Please see the back-gating data below.

new reviewer comment: The back-gating data was not included in the submission (or was not visible to the reviewer), therefore could not be verified. It is suggested that a supplemental figure with wider gating showing the live vs. dead population be added.

3. Figure resolution should be addressed upon final figure submission to ensure readability.

7. PLOS authors have the option to publish the peer review history of their article (what does this mean?). If published, this will include your full peer review and any attached files.

Reviewer #3: No

---

## [Author Response · Author response to Decision Letter 2]

26 Sep 2023

Characterization of polarization states of canine monocyte derived macrophages.

Qingkang Lyu, Edwin J.A. Veldhuizen , Irene S. Ludwig, Victor P. M. G. Rutten, Willem van Eden, Alice J. A. M. Sijts and Femke Broere 

Point to point reply. 

Comments to the Author

Reviewer #3: Overall, the authors have thoroughly addressed the majority of the reviewer comments. However, there are still a few items that require further clarification.

1. New reviewer comment: It’s the reviewer’s understanding that based on the manuscript changes the authors meant “…read counts lower than 2 in all samples…” rather than “2 fold difference”, please affirm whether this is true. If 2 fold difference was intended, please edit the manuscript accordingly.

We confirm that this is true. Genes with <2 counts for all samples, were not taken along in the determination which genes were differentially expressed. No changes made to the manuscript.

2. new reviewer comment: The back-gating data was not included in the submission (or was not visible to the reviewer), therefore could not be verified. It is suggested that a supplemental figure with wider gating showing the live vs. dead population be added.

We have now added the figure showing back-gating as a supplementary file (Fig S1) and referred to it in the text in Line 275 of the revised manuscript.

3. Figure resolution should be addressed upon final figure submission to ensure readability.

We checked figure resolution of all figures and they are all >300 dpi.

---

## [Decision Letter · Decision Letter 3]

28 Sep 2023

Characterization of polarization states of canine monocyte derived macrophages.

PONE-D-23-02588R3

Dear Dr. Veldhuizen,

We’re pleased to inform you that your manuscript has been judged scientifically suitable for publication and will be formally accepted for publication once it meets all outstanding technical requirements.

Kind regards,

Kenji Fujiwara, PhD, MD

Academic Editor

PLOS ONE

Additional Editor Comments (optional):

Dear Dr. Veldhuizen.

Thank you for revising your manuscript appropriately. All reviewers and I agreed to the acceptance.

Yours sincerely,

Kenji Fujiwara

Academic editor

Reviewers' comments:

Reviewer's Responses to Questions

**Comments to the Author**

1. If the authors have adequately addressed your comments raised in a previous round of review and you feel that this manuscript is now acceptable for publication, you may indicate that here to bypass the “Comments to the Author” section, enter your conflict of interest statement in the “Confidential to Editor” section, and submit your "Accept" recommendation.

Reviewer #3: All comments have been addressed

2. Is the manuscript technically sound, and do the data support the conclusions?

Reviewer #3: Yes

3. Has the statistical analysis been performed appropriately and rigorously? 

Reviewer #3: Yes

4. Have the authors made all data underlying the findings in their manuscript fully available?

Reviewer #3: Yes

5. Is the manuscript presented in an intelligible fashion and written in standard English?

Reviewer #3: Yes

6. Review Comments to the Author

Reviewer #3: (No Response)

7. PLOS authors have the option to publish the peer review history of their article (what does this mean?). If published, this will include your full peer review and any attached files.

Reviewer #3: No

---

## [Editor Report · Acceptance letter]

31 Oct 2023

PONE-D-23-02588R3 

Characterization of polarization states of canine monocyte derived macrophages. 

Dear Dr. Veldhuizen:

I'm pleased to inform you that your manuscript has been deemed suitable for publication in PLOS ONE. Congratulations! Your manuscript is now with our production department. 

Kind regards, 

on behalf of

Dr. Kenji Fujiwara 

Academic Editor

PLOS ONE